Study on Critical Rainfall for Flash Flood Disasters in Small 1 Watersheds of the Three Gorges Reservoir Area: A Case Study of 2 **Futian Small Watershed in Wushan County of Chongging** 3 4 Qu Guo<sup>1</sup>, Qin Yang<sup>1\*</sup>, Jun Kang<sup>1\*</sup>, Yi Liu<sup>2</sup>, Baogang Yang<sup>1</sup>, Huigen He<sup>1</sup>, Chuan Liu<sup>1</sup>, 5 Wanhong Gao<sup>3</sup> 6 <sup>1</sup>CMA Key Open Laboratory of Transforming Climate Resources to Economy, Yubei 401147, Chongqing, China 7 <sup>2</sup>CMA Chongqing Meteorological Observatory, Chongqing Meteorological Bureau, Chongqing, China 8 <sup>3</sup>Chongqing Wushan County Meteorological Bureau, Wushan 404700, Chongqing, China 9 10 \*First Corresponding Author: Qin Yang, CMA Key Open Laboratory of Transforming Climate Resources to 11 Economy, Yubei 401147, Chongqing, China, yqgugu2006@163.com; 12 \*Second Corresponding Author: Jun Kang, CMA Key Open Laboratory of Transforming Climate Resources to 13 Economy, Yubei 401147, Chongqing, China, 14 15 Abstract 16 Taking the Futian Small Watershed in Wushan, within the Three Gorges Reservoir area, as the research 17 object, this study utilized hourly rainfall data from 2010 to 2023 collected at the Futian Small 18 Watershed and nearby rainfall stations, historical disaster information on mountain flood disaster 19 processes, digital elevation models, land use data, and other relevant information. Statistical analysis 20 methods such as the single-station critical rainfall method, regional critical rainfall method, probability 21 distribution method, and the hydrodynamic model FloodArea, were employed to simulate and calculate 22 the critical rainfall amounts leading to disasters. Results indicate that the trends of critical rainfall 23 amounts leading to disasters calculated by various methods are generally consistent. However, at 24 different time scales, the critical rainfall amounts calculated by different methods exhibit variations. 25 The FloodArea simulation yields the smallest critical rainfall amounts for 1-hour, 2-hour, 24-hour 26 durations; the single-station critical rainfall method provides the smallest values for 5-hour, 6-hour, and 27 12-hour durations; the regional critical rainfall method gives the smallest results for 3-hour, 4-hour 28 durations. Statistical methods can swiftly and efficiently establish critical rainfall amounts leading to 29 disasters at different time scales, the FloodArea model can more precisely depict the 30 precipitation-runoff processes of mountain flood disasters. Therefore, by integrating statistical methods 31 with hydrological model simulations to leverage their respective strengths, we can more accurately 32 determine the critical rainfall amounts leading to mountain flood disasters. 33 Keywords: small watershed; flash flood; critical rainfall threshold; statistical analysis; FloodArea 34 1. Introduction 35 A small watershed typically refers to a relatively independent and enclosed natural catchment area 36 bounded by watershed divides and the outlet cross-section of downstream river channel below the 37 second or third-order tributaries, with a catchment area of less than 50 km? Small watersheds are 38 characterized by limited river channel storage capacity, steep slopes, short-duration floods, rapid rises 39 in water levels, and high flood peaks. Given the relatively low flood control standards of most small 40 and medium-sized rivers in China, floods in these rivers triggered by heavy rainstorms account for

https://doi.org/10.5194/egusphere-2025-3833 Preprint. Discussion started: 15 October 2025 © Author(s) 2025. CC BY 4.0 License.

70–80% of the total nationwide flood-related losses and 60–80% of the total flood-related fatalities (Tu et al., 2020). Therefore, implementing timely and effective meteorological service measures to mitigate or prevent flash flood disasters in small watersheds represents a crucial approach to meteorological disaster prevention and mitigation.

To mitigate and avert flash flood disasters in small watersheds, numerous scholars have conducted interdisciplinary research focusing on the formation mechanisms and impacts of such disasters, monitoring and early warning systems, risk assessment and zoning, as well as governance and mitigation strategies (Martina et al, 2006; Montesarchio et al, 2009; Merz et al, 2010; Fan et al., 2012; Zhang et al., 2012; Theule et al, 2012; Peng et al., 2017; Wang et al., 2024; Li et al., 2024). Among these, the critical rainfall threshold for disaster initiation serves as a pivotal indicator for flash flood forecasting and prediction, functioning as an "alarm trigger" within early warning systems. It denotes the rainfall amount that, if exceeded within a specific time window, will cause water depths at vulnerable points or critical infrastructure within the watershed to surpass warning levels, thereby triggering catastrophic flooding. Currently, two primary methodologies are employed to determine the critical rainfall thresholds for flash floods: statistical analysis and hydrological modeling (Lu et al., 2016; Zhao et al., 2019; Teng et al., 2023; Ewelina et al., 2023). The statistical analysis approach emphasizes quantifying mathematical correlations between historical flood parameters (e.g., inundation depths, discharges, water levels) and corresponding rainfall amounts, thereby inferring critical rainfall thresholds that may precipitate disasters. In contrast, the hydrological modeling approach calculates critical rainfall thresholds by simulating the physical processes governing storm-induced flood formation, including rainfall-runoff transformation, channel routing, and floodplain inundation dynamics.

Research on the critical rainfall thresholds triggering flash floods in small watersheds of the Three Gorges Reservoir Area (TGRA) remains limited. This region exhibits complex topography with diverse landforms, and its meteorological-hydrological conditions reflect dual characteristics of both reservoir zones and mountainous areas, resulting in flash flood formation mechanisms that significantly differ from those in typical mountainous or plain regions. In recent years, influenced by global climate change, extreme rainfall events have become increasingly frequent in the TGRA, with multiple flash flood disasters occurring in small watersheds during 2008, 2011, 2016, 2018, and 2023. These events have posed severe threats to human lives, property, and infrastructure. Additionally, as the world's largest hydraulic project, the safe operation of the Three Gorges Dam necessitates integrated risk management of flash floods in small watersheds within the reservoir area. Therefore, conducting analyses on critical rainfall thresholds for flash floods in TGRA small watersheds to enhance risk early warning services is a topic worthy of urgent exploration.

This study takes the Futian Small Watershed in Wushan County, Chongqing Municipality, as a case study. Leveraging historical disaster data from flash flood events in Futian Small Watershed spanning 2010–2023, along with hourly precipitation observations from regional automatic weather stations, we employ four methodologies—including hydrodynamic simulations using the FloodArea model, the single-station critical rainfall method, the regional critical rainfall method, and the probability distribution method—to calculate and determine critical rainfall thresholds for flash floods. By conducting a comparative analysis of critical rainfall values corresponding to different flash flood grades derived from these methods, we aim to provide a scientifically robust framework for determining flash flood critical rainfall thresholds and offer actionable references for the development

of storm-flood meteorological disaster risk early warning systems, risk zoning, and disaster prevention and mitigation strategies in the Three Gorges Reservoir Area (TGRA).

# 2. Study Area, Data and Methods

#### 2.1 Study Outline

The study area is located within Futian Town, Wushan County, Chongqing Municipality, situated in the core region of the Three Gorges Reservoir Area (Fig. 1) and classified as a sub-basin of the Daning River's Lesser Three Gorges watershed. This basin spans 54.57 km? with elevations ranging from 242 m to 1,568 m. The terrain along the flash flood gullies is characterized by hilly and mountainous landforms, exhibiting a topographic pattern of low-lying central valleys flanked by high-elevation slopes, with a maximum gradient reaching 72.3°. As a typical agricultural small watershed in the Three Gorges Reservoir Area, land use is predominantly arable (84.1%), concentrated along river corridors and valley floors, followed by grassland (7.7%). Residential areas are primarily clustered in the northeastern part of the basin (Fig. 2a). The basin experiences a humid climate, with an annual average precipitation of 1,074.1 mm, concentrated between May and September (accounting for ~70% of annual rainfall). Key vulnerability zones include the Futian Town government headquarters and market area, which are situated in densely populated regions with extensive arable land.

Under no-precipitation conditions, the flash flood gullies in this basin typically exhibit minimal standing water. However, during intense rainfall events, water depths in the gullies rise rapidly, increasing susceptibility to flash flood disasters. To mitigate flash flood risks, 2m-high embankments have been constructed along both sides of the gullies near critical vulnerability zones, designed to meet a 5-year return period flood protection standard. The basin is equipped with nine automatic rainfall observation stations, providing data sufficient for runoff simulation using the FloodArea hydrodynamic model. This makes the basin a representative case study area for analyzing critical rainfall thresholds for flash flood triggering in the Three Gorges Reservoir Area.

**Figure 1.** Overview Map of Futian Small Watershed. The small map in the top-left corner shows the boundary map of various districts and counties in Chongqing, with the blue lines representing the

Yangtze River and the light purple area indicating Wushan County; the small map in the top-right corner focuses on Wushan County, with the yellow area representing the Futian Small Watershed; the main map in the middle displays the detailed scope of the Futian Small Watershed, where the light blue lines are the mountain flood ditches of the small watershed extracted using ArcGIS, the red five-pointed stars mark the hazard points, and the light green dots represent rainfall stations.

#### 2.2 Data Sets

Historical disaster records of 19 flash flood events in the Futian Small Watershed were sourced from the Chongqing Municipal Leading Group Office for the First National Comprehensive Survey of Natural Disaster Risks and Wushan County's historical annals. Each record underwent rigorous field verification and validation, ensuring its status as valid sample data (Table 1). Hourly precipitation observations from nine rain gauge stations surrounding the Futian Small Watershed, provided by the Chongqing Meteorological Information Technology Support Center after passing homogeneity tests for data quality control.30 m-resolution Digital Elevation Model (DEM) data from the Shuttle Radar Topography Mission (SRTM), acquired through the National Aeronautics and Space Administration (NASA).Land cover data with a 30m spatial resolution was sourced from the GlobeLand30 official website.

**Table 1.** Historical Disaster Situations of Mountain Torrent Disasters in Futian Small Watershed from 2011 to 2023.

| Start T    | ime   | End Ti     | End Time |    | Disaster Situation                                                                                                                                                                                |
|------------|-------|------------|----------|----|---------------------------------------------------------------------------------------------------------------------------------------------------------------------------------------------------|
| 2011-08-04 | 17:00 | 2011-08-06 | 2:00     | 34 | The number of disaster-affected people is 7; 12 houses were damaged; the direct economic loss amounted to 200,000 yuan.                                                                           |
| 2013-05-08 | 23:00 | 2013-05-09 | 9:00     | 11 | The number of disaster-affected people is 3; the area of crops affected by the disaster is 11 hectares; the direct economic loss amounts to 500,000 yuan.                                         |
| 2014-08-31 | 07:00 | 2014-08-31 | 15:00    | 9  | The number of disaster-affected individuals is 1; three houses were damaged; the direct economic loss amounts to 36,000 yuan.                                                                     |
| 2015-05-14 | 20:00 | 2015-05-15 | 2:00     | 7  | The number of disaster-affected individuals is 1; one house was damaged.                                                                                                                          |
| 2016-06-01 | 16:00 | 2016-06-02 | 2:00     | 11 | The number of disaster-affected individuals is 6.                                                                                                                                                 |
| 2017-07-14 | 8:00  | 2017-07-14 | 19:00    | 12 | The area of crops affected by the disaster is 31 hectares; the direct economic loss amounts to 430,000 yuan.                                                                                      |
| 2017-08-08 | 1:00  | 2017-08-08 | 14:00    | 14 | The area of crops affected by the disaster is 4.7 hectares; one kilometer of the highway was damaged; the direct economic loss amounts to 370,000 yuan.                                           |
| 2018-06-18 | 18:00 | 2018-06-19 | 2:00     | 9  | The number of disaster-affected individuals is 18; the area of crops affected by the disaster is 113.33 hectares; six houses were damaged; the direct economic loss amounts to 5.22 million yuan. |
| 2020-07-16 | 6:00  | 2020-07-16 | 21:00    | 16 | The number of disaster-affected individuals is 6; two houses were damaged; the direct economic loss amounts to 10,000 yuan.                                                                       |
| 2021-08-10 | 22:00 | 2021-08-11 | 21:00    | 24 | The number of disaster-affected individuals is 1; the area of crops affected by the disaster is 1.03 hectares; the direct economic loss amounts to 232,000 yuan.                                  |

|            |       |            |       |    | The number of disaster-affected individuals is 11; the area of crops affected by the |
|------------|-------|------------|-------|----|--------------------------------------------------------------------------------------|
| 2021-08-25 | 23:00 | 2021-08-27 | 14:00 | 40 | disaster is 0.47 hectares, and three houses                                          |
|            |       |            |       |    | were damaged; the direct economic loss                                               |
|            |       |            |       |    | amounts to 716,800 yuan.                                                             |
|            |       |            |       |    | The number of disaster-affected individuals                                          |
| 2021-09-19 | 4:00  | 2021-09-19 | 13:00 | 10 | is 1; the direct economic loss amounts to                                            |
|            |       |            |       |    | 150,000 yuan.                                                                        |
|            | 42.00 | 2022 04 42 |       |    | The number of disaster-affected individuals                                          |
| 2022-04-12 | 12:00 | 2022-04-12 | 22:00 | 11 | is 3; the direct economic loss amounts to                                            |
|            |       |            |       |    | 30,000 yuan.                                                                         |
| 2022-06-27 | 3:00  | 2022-06-27 | 16:00 | 14 | The number of disaster-affected individuals                                          |
|            |       |            |       |    | is 7. The number of disaster-affected individuals                                    |
| 2022-10-05 | 22:00 | 2022-10-06 | 18:00 | 21 |                                                                                      |
| 2022-10-03 | 22.00 | 2022-10-00 | 18.00 | 21 | is 6; two houses were damaged; the direct economic loss amounts to 6,000 yuan.       |
|            |       |            |       |    | The area of crops affected by the disaster is                                        |
| 2023-05-04 | 7:00  | 2023-05-04 | 9:00  | 3  | 1.99 hectares: the direct economic loss                                              |
| 2023-03-04 | 7.00  | 2023-03-04 | 9.00  | 3  | amounts to 19,500 yuan.                                                              |
|            |       |            |       |    | The area of crops affected by the disaster is                                        |
| 2023-06-18 | 6:00  | 2023-06-18 | 11:00 | 6  | 1.6 hectares; the direct economic loss                                               |
| 2020 00 10 | 0.00  | 2020 00 10 | 11.00 | Ü  | amounts to 8,800 yuan.                                                               |
|            |       |            |       |    | The number of disaster-affected individuals                                          |
|            |       |            |       |    | is 8; the area of crops affected by the                                              |
| 2023-06-29 | 2:00  | 2023-06-29 | 21:00 | 20 | disaster is 2.1 hectares; the direct economic                                        |
|            |       |            |       |    | loss amounts to 15,800 yuan.                                                         |
|            |       |            |       |    | The number of disaster-affected individuals                                          |
| 2023-07-04 | 0:00  | 2023-07-04 | 11:00 | 12 | is 1; one house was damaged; the direct                                              |
|            |       |            |       |    | economic loss amounts to 15,000 yuan.                                                |

# 2.3 Methods

## 2.3.1 Extraction of Flash Flood Gullies

The extraction of flash flood gullies and their boundaries within the Futian Small Watershed (Fig. 1) was conducted using ArcGIS 10.2 (hereafter referred to as GIS) through the following steps: sink filling, flow direction generation, flow accumulation calculation, stream network extraction, watershed delineation, and sub-basin merging. GIS-derived analysis identified the longest flash flood gully extending to the vulnerability zone as 10,356 meters in length. Using the formula "Concentration Time = Length  $\times 0.012 / 1000$ " (Length refers to the length of the flash flood gully) (Liu et al., 2021), the concentration time was calculated as 2.98 hours.

## 2.3.2 Areal Precipitation Calculation

Areal precipitation for the 19 flash flood events in the Futian Small Watershed was computed using Thiessen polygons generated from precipitation observations at nine rain gauge stations surrounding the basin and GIS-based spatial analysis tools. The area-weighted method was applied to calculate basin-wide precipitation, with detailed computational procedures outlined in references (Tang ,2012).

# 2.3.3 Surface Hydraulic Roughness and Surface Runoff Coefficient

Surface hydraulic roughness is a composite coefficient that quantifies the effects of irregularities and roughness on the walls of river channels or gullies, primarily reflecting hydraulic friction in fluid dynamics. Using land cover classifications within the Futian Small Watershed, initial values for the Strickler coefficient (K)—a measure of surface hydraulic roughness—were assigned based on research by Zhang et al. (1994) and Tong et al. (2011). The Soil Conservation Service (SCS) Curve Number (CN) model, a land-use-based parameterization approach, objectively captures the impacts of soil type, land use practices, and antecedent soil moisture on rainfall-runoff relationships, enabling accurate estimation of watershed-scale runoff (Gan et al., 2010). By applying land cover

classifications, runoff coefficient values were computed for each flash flood event in the Futian Small Watershed. Figure 2b illustrates the spatial distribution of runoff coefficients during the flash flood event of June 18, 2023.

#### 2.3.4 FloodArea Model

The FloodArea model (hereafter referred to as FA) is a GIS-integrated flood inundation simulation system developed by Geomer GmbH (Germany). The model simulates flood inundation by integrating key influencing factors such as topographic characteristics, rainfall intensity, and spatial distribution, utilizing input precipitation data and Digital Elevation Model (DEM) datasets. Technical details of the model's methodology are documented in Geomer (2003). The model demonstrates superior performance in mountainous regions (Xu et al., 2024; Wei et al., 2024). Given the absence of hydrological stations and historical records of water levels/discharges in most flash flood gullies within the Three Gorges Reservoir Area (TGRA), the FloodArea model is particularly suitable for determining critical rainfall thresholds for flood-induced disasters in such data-scarce basins (Ji et al., 2015).

**Figure 2.** Land use Map of Futian Small Watershed (a) and Distribution Map of Surface Runoff Coefficients during the Mountain Flood Disaster Process on June 18, 2023 (b).

## 3. Results

# 3.1 FloodArea (FA) Model Simulations

Based on 19 historical flash flood events recorded in the Futian Small Watershed between 2011 and 2023, 10 events (occurring on August 4, 2011; May 8, 2013; June 1, 2016; July 14, 2017; August 8, 2017; June 18, 2018; July 16, 2020; August 25, 2021; September 19, 2021; and June 18, 2023) were randomly selected for analysis. Areal precipitation data for each event, along with Digital Elevation Model (DEM) datasets, surface runoff coefficients, and surface hydraulic roughness, were input into the FloodArea (FA) model to simulate flood inundation dynamics. Simulations were conducted with a 1-hour temporal resolution.

Figure 3 shows the FA simulated submergence depth change by taking the mountain torrent disaster process on June 18, 2023 as an example. It can be seen that at 6:00 on June 18, there was a sudden heavy rainfall in the basin. At 6:00, the area rainfall in the basin reached 8.7mm. Only a small amount of water accumulated in the upstream area, and the remaining mountain torrents have not yet formed an

208

effective submerged depth (Fig. 3a). The continuous heavy rainfall from 7:00 to 9:00 led to the rapid rise of the submerged depth of the Shanhong gully. At 9:00, the maximum submerged depth of the flood occurred successively at each section (Fig. 3b). From 9:00 to 10:00, the precipitation magnitude decreased significantly compared with the previous period, and the precipitation process ended at 11:00; At 11 o'clock, due to the change of confluence conditions in the upstream area of shanhonggou, the submerged depth had dropped to zero, while the downstream area was affected by the regulation and storage of the river channel, and the submerged depth was maintained at a level similar to that at 9 o'clock (Fig. 3C). At 14:00, as the drainage process of the basin continued, the flood of shanhonggou gradually subsided and the submerged area significantly contracted (Fig. 3D). From the perspective of the whole simulation process, almost all of the middle and lower reaches of Shanhong gully in Futian small watershed have been flooded to varying degrees, and the submerged depth in many places is more than 3 m; this process is characterized by strong sudden precipitation, rapid formation of flood peak, and long duration of submerged depth in the middle and lower reaches due to river regulation and storage. The simulation results of this process are in good agreement with the actual situation. According to the field survey and review, about 1.6 hectares of crops near the hidden danger point were affected, and the direct economic loss was 8800 yuan. Next, based on the results of each mountain flood disaster process simulated by FA, the hourly inundation depth of the hidden danger point is extracted, and then the equation between the cumulative area rainfall of different time lengths such as 1, 2, 3, 4, 5, 6, 12 and 24h and the inundation depth is established respectively. Finally, the inundation depth of different mountain flood risk warning levels is substituted into the equation to obtain the corresponding disaster causing critical area rainfall under different mountain flood levels.

**Figure 3.** Changes in the inundation depths of the mountain flood process in Futian Small Watershed simulated by FloodArea at 6:00 (a), 9:00 (b), 11:00 (c), and 14:00 (d) on June 18, 2023. The varying

shades of blue in the figure represent different inundation depths, while the varying shades of gray indicate different elevations.

Figure 4 shows the relationship between the submerged depth of hidden danger points and the process area rainfall It can be seen from the figure that the ratio of peak period and duration of area rainfall during 10 mountain torrents is 0.3 on average, and the peak rainfall accounts for an average of 23.7% of the process rainfall. The process area rainfall shows the distribution characteristics that the precipitation in the previous period is large, and the rainfall decreases with time; The 8/10 mountain torrent process is single peak type, with less bimodal or multi peak type, and the 7/10 single peak rainfall is concentrated, which is easy to form rainwater convergence in a short time. The area rainfall and inundation depth of each mountain torrent process respond well, and the occurrence time of the peak of inundation depth lags behind the peak of area rainfall, with a lag time of about 1-3H. The main gully of the mountain torrent gully in the small watershed of Hilly and mountainous areas is short and has a large gradient. When the rainstorm strikes, the flood converges rapidly, and it usually takes only 1-2h from the beginning of rainfall to the occurrence of mountain torrents (Hapuarachchi et al., 2011; Wang, 2011). For example, from 07:00 to 09:00 on May 4, 2023, the precipitation was only 3 hours, the area rainfall reached 34.9mm, and the maximum area rainfall reached 19.6mm, causing 1.99 hectares of crops to be affected, and the direct economic loss was 19500 yuan.

**Figure 4.** Diagram of Areal Precipitation and Inundation Depth Changes During Each Flash Flood Disaster Process at Hazard Points in Futian Small Watershed. The blue bars represent the areal precipitation of the small watershed, while the orange curve indicates the inundation depth at the hazard points.

There is a corresponding relationship between the submerged water depth and the amount of precipitation. Using FA simulated data of submerged water depth at hidden points in the process of mountain torrents in Futian small watershed and accumulated area rainfall at different time scales, the relationship between the two is scattered and generalized into a single relationship curve (Fig. 5). It can be seen that the correlation coefficients between cumulative precipitation and submerged depth of hidden danger points at different time scales are more than 0.7, which have passed the significance test of 0.01, and the correlation coefficient of cumulative area rainfall at 12 h and 6 h is the highest, about 0.9. Therefore, the time scales of 1, 2, 3, 4, 5, 6, 12 and 24 hours can be selected to determine the rainfall at the disaster causing critical interface in Futian small watershed.

**Figure 5.** Diagram showing the relationship between inundation depth and cumulative areal precipitation at hazard points in Futian Small Watershed, as simulated by FA, for time intervals of 1 hour (a), 2 hours (b), 3 hours (c), 4 hours (d), 5 hours (e), 6 hours (f), 12 hours (g), and 24 hours (h). The black dots represent the inundation depths corresponding to the cumulative areal precipitation, while the black dashed lines indicate the binomial fitting trend lines.

The process area rainfall and submerged depth were determined by FA simulation. The regression equations of area rainfall and submerged depth of hidden danger points at different time scales were as follows:

| 257 | $y_{1h} = 0.5971x^2 + 2.9667x + 4.2534$      |
|-----|----------------------------------------------|
| 258 | $y_{2h} = 0.2573x^2 + 5.6321x + 7.6453$      |
| 259 | $y_{3h} = -0.4211x^2 + 11.25x + 9.0727$      |
| 260 | $y_{4h} = -1.5082x^2 + 17.043x + 9.9831$     |
| 261 | $y_{5h}$ = -1.1267 $x^2$ +19.281 $x$ +9.6516 |
| 262 | $y_{6h} = -2.3574x^2 + 26.904x + 8.6576$     |
| 263 | $y_{12h} = -0.4888x^2 + 22.831x + 11.363$    |
| 264 | $y_{24h} = -1.5991x^2 + 29.279x + 15.914$    |

In the formula, X is the submergence depth of the hidden danger point, and Y is the accumulated area rainfall at different time scales. According to the technical guide for determining the critical (area) rainfall caused by rainstorm and flood disasters issued by the China Meteorological Administration, the impact of flood inundation depth on people can be divided into three risk levels: 0.6 m for level III risk, 1.2m for level II risk, and 1.8 m for level I risk (Yu et al., 2017). Due to the construction of 2-meter-high dikes on both sides of the flash flood gully near the hidden danger point, the actual inundation depth is adjusted to 2.6m (level III), 3.2m (level II) and 3.8m (level I) for analysis when calculating the critical rainfall threshold of flash flood warning level.

Table 2 shows the FA simulated rainfall at the hazard critical interface of three early warning levels at different time scales of hidden danger points in Futian small watershed. It can be seen that under the same warning level, the shorter the time, the lower the disaster critical rainfall. This means that in a short period of time, less precipitation can lead to the same disaster risk as a longer period of strong precipitation. For example, 16.1mm precipitation in 1h, 62.7mm precipitation in 6h, and 81.2mm precipitation in 24h may raise the submerged depth of hidden danger points to 2.6m. From the perspective of hydrology, when the rainfall intensity per unit time increases, the precipitation will rapidly exceed the infiltration capacity of the underlying surface; In this case, precipitation cannot be completely infiltrated, but directly transformed into surface runoff. This phenomenon reduces the consumption of rainfall infiltration, leading to the reduction of rainfall at the critical interface, which can explain and verify the theory of excess infiltration runoff in hydrology. In addition, the higher the warning level and the longer the cumulative time, the greater the critical rainfall. Within 6 hours, the required precipitation from the third level early warning to the second level early warning is 70.6mm, and the required precipitation to reach the first level early warning is 76.9mm.

**Table 2.** Critical Rainfall Amounts Simulated by FA for Hazard Points in Futian Small Watershed at Different Inundation Depths (level I: 3.8 m, level II: 3.2 m, level III: 2.6 m) for Durations of 1h, 2h, 3h, 4h, 5h, 6h, 12h, and 24h.

| Warning   | Inundation |      |      |      | Time | Scale |      |      |       |
|-----------|------------|------|------|------|------|-------|------|------|-------|
| Level     | Depth/m    | 1h   | 2h   | 3h   | 4h   | 5h    | 6h   | 12h  | 24h   |
| Level III | 2.6        | 16.1 | 24.1 | 35.5 | 44.1 | 52.2  | 62.7 | 67.4 | 81.2  |
| Level II  | 3.2        | 19.9 | 28.3 | 40.8 | 49.1 | 59.8  | 70.6 | 79.4 | 93.2  |
| Level I   | 3.8        | 24.2 | 32.8 | 45.7 | 53.0 | 66.6  | 76.9 | 91.1 | 104.1 |

### 3.2 Statistical Analysis

## 3.2.1 Single Station Critical Rainfall Method

The single station critical rainfall method is based on the hourly rainfall and calculates the maximum rainfall of each rainfall station at different time scales during the historical mountain torrent disaster process in a small watershed by hourly moving average. Because the rainfall stations around Futian small watershed are unevenly distributed and sparse, the average of the maximum rainfall of all rainfall stations in the watershed is selected as the critical rainfall value at different time scales. According to the regulations of the China Meteorological Administration, rain with rainfall of more than 16mm in

one hour, or more than 30mm in 12 consecutive hours, and 50mm or more in 24 hours is called "rainstorm". Table 3 shows the calculation results of critical rainfall at different time scales for each rainfall station during the historical mountain torrent disaster process in Futian small watershed. It can be seen that each rainfall station has reached the rainstorm level within 1h, the precipitation is 17.5-25.8mm, the difference between the maximum precipitation ( $R_{max}$ ) and the minimum precipitation ( $R_{min}$ ) is 8.3mm, and the average precipitation ( $R_{ave}$ ) is 21.3mm; the precipitation in 2h is 24.2-36.8 mm, the difference between  $R_{max}$  and  $R_{min}$  is 12.6mm,  $R_{ave}$  is 36.1mm; the precipitation in 3h is 29.2-43.7mm,  $R_{ave}$  is 36.1mm; the precipitation in 4h is 34.7-51.9mm,  $R_{ave}$  is 42.6mm; the precipitation in 5h is 38.6-60.  $R_{ave}$  is 47.9mm; The precipitation in 6h is 42.9-64.0 mm,  $R_{ave}$  is 52.3mm; The maximum rainfall value of each station in 12h is greater than the rainstorm level,  $R_{ave}$  is 66.9mm; the maximum rainfall value in 24h is greater than the rainstorm level,  $R_{ave}$  is 86.8mm.

**Table 3.** Precipitation at different time scales for various rainfall stations during historical flash flood disaster processes in Futian Small Watershed.

| Station Name     | Station |      |      |      | Time | Scale |      |      |       |
|------------------|---------|------|------|------|------|-------|------|------|-------|
| Station Name     | Number  | 1h   | 2h   | 3h   | 4h   | 5h    | 6h   | 12h  | 24h   |
| Futian           | A7204   | 19.4 | 27.6 | 33.4 | 38.3 | 43.4  | 46.7 | 62.5 | 90.5  |
| Longwang Cun     | A7994   | 25.8 | 36.8 | 43.7 | 51.9 | 60.4  | 64.0 | 82.2 | 91.6  |
| Laoya Cun        | A7995   | 21.4 | 30.0 | 38.8 | 45.2 | 50.7  | 54.0 | 71.1 | 103.0 |
| Tiangong Cun     | A8015   | 17.5 | 24.2 | 29.2 | 34.7 | 38.6  | 42.9 | 57.6 | 78.6  |
| Shangtan         | A8372   | 20.9 | 28.9 | 35.0 | 41.0 | 46.0  | 50.1 | 64.5 | 70.6  |
| Tianchi          | A8377   | 24.4 | 32.9 | 39.7 | 46.4 | 51.0  | 59.1 | 75.0 | 84.4  |
| Qifeng           | A8541   | 23.9 | 31.9 | 37.8 | 46.0 | 51.7  | 55.3 | 65.7 | 97.6  |
| Wulong Cun       | A8823   | 19.8 | 29.5 | 35.4 | 42.3 | 48.1  | 52.2 | 65.5 | 76.9  |
| Longwu Cun       | A8824   | 18.9 | 27.0 | 32.1 | 37.2 | 41.1  | 46.5 | 58.1 | 87.7  |
| R <sub>max</sub> |         | 25.8 | 36.8 | 43.7 | 51.9 | 60.4  | 64.0 | 82.2 | 103.0 |
| $R_{min}$        |         | 17.5 | 24.2 | 29.2 | 34.7 | 38.6  | 42.9 | 57.6 | 70.6  |
| Rave             |         | 21.3 | 29.9 | 36.1 | 42.6 | 47.9  | 52.3 | 66.9 | 86.8  |

# 3.2.2 Regional Critical Rainfall Method

The regional critical rainfall method is based on the area rainfall of the basin, not on the rainfall of a single station. Its value reflects the average state of rainfall in the basin. This method is more suitable for areas with sparse rainfall stations. Table 4 shows the regional critical rainfall at different time scales of the mountain torrent disaster process in Futian small watershed. It can be seen that the critical rainfall in 1h region is between 12.9-36.5mm,  $R_{ave}$  is 19.8mm; the critical rainfall in 2h region is between 19.1-58.1mm,  $R_{ave}$  is 28.6mm; the critical rainfall in 3h region is between 21.9-68.9mm,  $R_{ave}$  is 34.9mm; the critical rainfall in 4h region is between 23.2-86.4mm,  $R_{ave}$  is 41.9mm; the critical rainfall in 5h region is between 24.5-101.9mm,  $R_{ave}$  is 48.0mm; the critical rainfall in 6h region is between 25.1-105.3mm,  $R_{ave}$  is 53.1mm; the critical rainfall in 12h area is 37.2-110.7mm,  $R_{ave}$  is 67.9mm; the critical rainfall in 24h area is 51.7-122.4mm,  $R_{ave}$  is 92.2mm..

**Table 4.** Areal Precipitation of the Watershed at Different Time Scales for Each Flash Flood Disaster Process in Futian Small Watershed.

|                                 | Duration                     |    |    |    | Tin | ne Scale |    |     |     |
|---------------------------------|------------------------------|----|----|----|-----|----------|----|-----|-----|
| Flash Flood<br>Disaster Process | of the<br>Process<br>(hours) | 1h | 2h | 3h | 4h  | 5h       | 6h | 12h | 24h |

| 2011-08-04 | 34 | 11.5 | 16.5 | 21.7 | 26.1 | 31.6  | 38.6  | 61.6  | 92.0  |
|------------|----|------|------|------|------|-------|-------|-------|-------|
| 2013-05-08 | 11 | 12.9 | 19.1 | 25.4 | 31.6 | 36.5  | 40.3  | 46.0  |       |
| 2016-06-01 | 11 | 16.4 | 21.3 | 25.4 | 29.0 | 33.5  | 38.6  | 55.4  |       |
| 2017-07-14 | 12 | 19.7 | 26.2 | 28.8 | 32.6 | 37.1  | 41.1  | 52.2  |       |
| 2017-08-08 | 14 | 31.4 | 42.4 | 51.2 | 65.9 | 75.2  | 85.4  | 96.0  | 102.7 |
| 2018-06-18 | 9  | 36.5 | 58.1 | 68.9 | 86.4 | 101.9 | 105.3 | 110.7 |       |
| 2020-07-16 | 16 | 14.0 | 20.3 | 21.9 | 23.2 | 24.5  | 25.1  | 37.2  | 51.7  |
| 2021-08-25 | 40 | 14.1 | 22.8 | 31.6 | 39.1 | 44.0  | 50.0  | 90.1  | 122.4 |
| 2021-09-19 | 10 | 23.6 | 32.3 | 40.9 | 45.9 | 51.7  | 57.6  | 62.1  |       |
| 2023-06-18 | 6  | 18.2 | 26.9 | 33.1 | 38.9 | 43.9  | 48.7  |       |       |
| $R_{max}$  |    | 36.5 | 58.1 | 68.9 | 86.4 | 101.9 | 105.3 | 110.7 | 122.4 |
| $R_{\min}$ |    | 11.5 | 16.5 | 21.7 | 23.2 | 24.5  | 25.1  | 37.2  | 51.7  |
| Rave       |    | 19.8 | 28.6 | 34.9 | 41.9 | 48.0  | 53.1  | 67.9  | 92.2  |

#### 3.2.3 Probability Distribution Method

The probability distribution method is based on the same frequency of mountain torrents and rainfall, and combined with the actual situation of local flood control projects to determine the critical rainfall. At present, the academia has not formed a unified standardized model for the probability distribution modeling of precipitation extreme value. In this paper, five probability distribution functions such as generalized extreme value distribution, Poisson distribution, lognormal distribution, exponential distribution and gamma distribution commonly used in the field of meteorology are selected, and the maximum likelihood estimation method is used to fit the rainfall extreme value series at different time scales, so as to obtain the initial value of critical rainfall. Table 5 shows the statistical results of the fitting errors of various distribution functions. Through the comparative analysis of K-S test values, it is found that the fitting effect of generalized extreme value distribution is the best, followed by gamma distribution and exponential distribution. Specifically, the fitting results of generalized extreme value distribution meet the requirements of 0.05 significance level, showing good goodness of fit, and can accurately fit the precipitation and its probability in Futian small watershed; In particular, the fitting accuracy of 2-hour and 12 hour time scale data is relatively higher, and the test results reach the significance level of 0.01.

**Table 5.** Goodness-of-fit test results (K-S test statistic values) for five probability models—
Generalized Extreme Value Distribution, Poisson Distribution, Lognormal Distribution, Exponential
Distribution, and Gamma Distribution—applied to rainfall extreme value series at different time scales.

| Time<br>Scale<br>/h | ① Generalized Extreme Value Distribution | ②Poisson<br>Distribution | ③<br>Lognormal<br>Distribution | ④<br>Exponential<br>Distribution | ⑤Gamma<br>Distribution | Sorting by K-S<br>Values |
|---------------------|------------------------------------------|--------------------------|--------------------------------|----------------------------------|------------------------|--------------------------|
| 1                   | 0.68                                     | 0.28                     | 0.73                           | 0.39                             | 0.69                   | 3>5>1>4>2                |
| 2                   | 0.98                                     | 0.31                     | 0.78                           | 0.36                             | 0.43                   | 1>3>5>4>2                |
| 3                   | 0.41                                     | 0.29                     | 0.22                           | 0.39                             | 0.36                   | 1>4>5>2>3                |
| 4                   | 0.77                                     | 0.33                     | 0.53                           | 0.34                             | 0.21                   | 1>3>4>2>5                |
| 5                   | 0.79                                     | 0.37                     | 0.34                           | 0.34                             | 0.11                   | 1>2>3>4>5                |
| 6                   | 0.38                                     | 0.32                     | 0.27                           | 0.37                             | 0.23                   | 1>4>2>3>5                |
| 12                  | 0.95                                     | 0.34                     | 0.18                           | 0.36                             | 0.63                   | 1>5>4>2>3                |
| 24                  | 0.70                                     | 0.32                     | 0.24                           | 0.36                             | 0.62                   | 1>5>4>2>3                |

The flood control capacity of the hidden danger points in the Futian small watershed is 5a/once-in-a-century, which means that the time interval between the occurrence of mountain flood disasters is approximately 5a. Taking the recurrence period of 5a as the critical rainfall for mountain floods calculated by the probability distribution method, the calculation results are shown in Table 6.

**Table 6.** Critical rainfall amounts with a 5-year return period at different time scales for hazard points in Futian Small Watershed, calculated based on the Generalized Extreme Value distribution function.

| Return Period /a |      |      |      | Tin  | ne Scale |      |      |      |
|------------------|------|------|------|------|----------|------|------|------|
| Return Period /a | 1 h  | 2h   | 3h   | 4h   | 5h       | 6h   | 12h  | 24h  |
| 5                | 27.9 | 30.0 | 45.0 | 47.5 | 49.0     | 59.9 | 76.7 | 93.4 |

A comparative analysis of the critical rainfall thresholds determined by the single-station critical rainfall method, the regional critical rainfall method, and the probability distribution method reveals that the results from the single-station critical rainfall method and the regional critical rainfall method are generally similar, but there are differences at different time scales. Among them, the calculation results of the critical rainfall method in the 1-4h region are slightly lower than those of the single-station method; However, the calculation results of the 5-24h regional method are higher than those of the single-station method. The calculation results of the probability distribution method are significantly higher than those of the two methods mentioned above, with critical rainfall values exceeding those of the single-station method and the regional method by approximately 11% to 14%.

# 3.3 Verification and Optimization

To verify the accuracy of the disaster-causing critical rainfall values determined by different methods, a comparative analysis was conducted using nine flash flood events that occurred between 2010 and 2023 and did not overlap with the aforementioned study (namely, on August 31, 2014; May 14, 2015; August 10, 2021; April 12, 2022; June 27, 2022; October 5, 2022; May 4, 2023; June 29, 2023; and July 4, 2023) (Figure 6). The results showed that the minimum values of the warning critical rainfall calculated by the four methods varied across different time scales: at the 1-hour, 2-hour, and 24-hour time scales, the critical rainfall values calculated by FA were the smallest; at the 3-hour and 4-hour time scales, the results from the regional critical rainfall method were the smallest; and at the 5-hour, 6-hour, and 12-hour time scales, the results from the single-station critical rainfall method were the smallest.

Further analysis revealed that, among these nine flash flood events, approximately 25% of the process rainfall amounts at each time scale were below the minimum critical values calculated by the four methods; however, in all flash flood events, there was at least one time scale where the rainfall exceeded the minimum critical values calculated by the four methods; when the rainfall exceeded this critical value, there was a potential for disaster.

Based on the above analysis, the minimum values calculated by the four methods at each time scale were set as the early warning indicators for flash floods triggered at that time scale (Table 7). When precipitation exceeds this indicator, the likelihood of a flash flood disaster increases. This setting not only retains the single-point extreme value characteristics of the single-station method but also takes

389

390

into account the spatial averaging effect of the regional method. Additionally, cross-validation ensures
 the rationality and reliability of the thresholds, enabling effective capture of potential flash flood risks.

**Figure 6.** Comparison of areal precipitation at different time scales (a: 1h, b: 2h, c: 3h, d: 4h, e: 5h, f: 6h, g: 12h, h: 24h) during nine flash flood disaster processes in Futian Small Watershed with critical rainfall values calculated using four methods based on data from another ten flash flood disaster processes.

Table 7. Disaster-causing Critical Rainfall Amounts at Different Time Scales (1h, 2h, 3h, 4h, 5h, 6h, 12h, and 24h) for Hazard Points in Futian Small Watershed. The minimum values calculated by four methods—single-station critical rainfall method, regional critical rainfall method, probability distribution method, and FloodArea simulation (FA)—at various time scales are used as indicators for disaster-causing critical rainfall triggering flash floods. Specifically, the FA calculation results are adopted for the 1h, 2h, and 24h time scales; the regional critical rainfall method results are used for the 3h and 4h time scales; and the single-station critical rainfall method results are applied for the 5h, 6h, and 12h time scales.

| Time Scale /h                      | 1h | 2h | 3h | 4h | 5h | 6h | 12h | 24h |
|------------------------------------|----|----|----|----|----|----|-----|-----|
| Critical Rainfall<br>Threshold /mm | 16 | 24 | 34 | 41 | 47 | 52 | 66  | 81  |

### 4. Conclusion and Discussion

FA can better invert the inundation process of historical mountain flood disasters in Futian small watershed, and the simulated inundation depth and surface rainfall response are good. The peak time of inundation depth lags behind the peak time of surface rainfall by about 1-3 hours. The precipitation peak of mountain flood disaster process is in the front and the rainfall is concentrated, which is likely to form flood confluence and cause mountain flood disasters in a short period of time.

Randomly select 10 processes from 19 historical mountain flood disasters in the Futian small watershed since 2011 for analysis using four methods: FA simulation, single-station critical rainfall method, regional critical rainfall method, and probability distribution method. The minimum critical rainfall values are obtained using FA simulation for 1h, 2h, and 24h, while the minimum results are obtained using the single-station critical rainfall method for 5h, 6h, and 12h, and the minimum results are obtained using the regional critical rainfall method for 3h and 4h. The critical rainfall at different time scales was tested and optimized using another 9 mountain flood disaster processes that did not participate in the analysis. At least one of the 9 mountain flood processes had a rainfall at a time scale greater than the minimum critical value of the 4 methods. The minimum values calculated by the 4 methods at each time scale were selected as the critical rainfall for mountain flood disasters in the Futian small watershed.

This article takes the Futian small watershed in the hinterland of the Three Gorges Reservoir area as an example, and compares four methods for determining critical rainfall. It is found that the critical rainfall method and the regional critical rainfall method mainly rely on local rainfall data conditions and historical records of mountain flood disasters. When the density of rainfall stations is high enough to cover each tributary of the small watershed, if the underlying surface conditions remain unchanged, the single-station critical rainfall method can basically achieve real-time monitoring and early warning for the small watershed. The regional critical rainfall method is more suitable for areas with sparse distribution of rainfall stations; The probability distribution method is relatively simple in terms of data requirements, requiring only the statistics of the years in which flash floods have occurred in the small watershed. It is suitable for areas where the disaster records are not complete and the distribution of stations is sparse. However, this method does not consider the situation where two or more disasters occur in a small watershed in a year. When the frequency of flash floods is low, there may be a problem of high critical rainfall values in the calculation results; The FA model is suitable for determining the critical rainfall for disasters in small mountainous watersheds without hydrological observation data, but the results vary greatly depending on the location of potential hazards, resulting in

different critical rainfall values. For areas with complete records of mountain torrents and rainfall data, it is recommended to comprehensively use multiple methods to review the critical rainfall for disaster causing, and improve the effectiveness of mountain torrent forecasting and early warning through inspection and optimization.

Due to limitations in data, technical methods, and other factors, this study also has certain limitations. Firstly, in the statistical calculation of rainfall characteristics, this article smoothes the actual precipitation situation to a certain extent, resulting in the inability to accurately restore the true distribution of the precipitation process in terms of temporal variation during the flash flood disaster. It also fails to fully reflect the spatial distribution characteristics of rainfall, such as whether precipitation is concentrated in the downstream or upstream, in terms of spatial variation; In practical work, the true spatial and temporal distribution of rainfall in a watershed can be obtained by analyzing the precipitation process and spatial distribution of rainfall patterns in small watersheds. Secondly, the limited number of hydrological observation stations in small watersheds is not conducive to accurately determining the critical rainfall threshold for disasters; More historical data is needed to simulate and verify the precipitation conditions in the same basin multiple times to enhance the representativeness of the results. Thirdly, in the FA simulation of rainstorm and mountain torrents, the simulation is mainly based on DEM, and the accuracy of DEM has a significant impact on the accuracy of threshold determination; The DEM accuracy used in this article is 30m ×30m, which is equivalent to the same altitude within this grid range, making it difficult to describe the uneven characteristics of the river channel in detail, resulting in a certain deviation in the threshold value for determining potential hazards. Finally, the occurrence of mountain flood disasters in small watersheds in the Three Gorges Reservoir area is affected by a variety of factors, including rainfall-induced disaster factors, characteristics of the underlying surface of the watershed, such as changes in water levels in the Three Gorges Reservoir area, and human activities such as disaster prevention and mitigation engineering facilities; The probability of extreme rainfall events varies significantly across different months or seasons within the same basin, which can lead to significant changes in the rainfall threshold at different time scales. The critical rainfall threshold for disaster prevention is not a fixed value, but rather a variable that is dynamically related to the aforementioned factors. In view of this, the next step of research will be based on refined underlying surface data, selecting different scales and types of small watersheds in the reservoir area, and using more methods such as machine learning and artificial intelligence to study the critical rainfall for mountain torrents disaster in small watersheds in the reservoir area, in order to provide more scientific warning thresholds for scientifically standardized monitoring of mountain torrents in small watersheds in the Three Gorges Reservoir area.

# Author contributions

- Qu Guo: Responsible for processing precipitation observation data from rainfall stations, as well as writing and revising the article.
- Qin Yang: Responsible for proofreading and processing disaster data, and polishing the article.
- Jun Kang: Responsible for conducting Floodarea simulations for various historical mountain flood
- disaster processes
- Yi Liu: Responsible for the verification and optimization of critical rainfall thresholds for disaster
- causation.

statistical analysis methods such as the single-station critical rainfall method, regional critical rainfall 474 method. 475 Huigen He:Responsible for calculating critical rainfall thresholds for disaster causation using statistical 476 analysis methods such as probability distribution method. 477 Chuan Liu: Responsible for writing the conclusions and discussions section of the article. 478 Wanhong Gao: Responsible for investigating and reviewing historical disaster data of mountain floods. 479 Competing interests 480 The contact author has declared that none of the authors has any competing interests. 481 Acknowledgments 482 This study was supported by the National Natural Science Foundation of China [Grant Numbers 483 41875111]; the Chongqing Natural Science Foundation General Project [Grant Numbers 484 CSTB2022NSCQ-MSX0558]; and the Chongqing Meteorological Department Business Technology 485 Research and Development Project [Grant Numbers YWJSGG-202305]. 486 Code/Data availability 487 Code/Data can be made available on request. 488 References 489 TU Yong., WU, Zebin., and HE Bingshun.: Analysis on the characteristics of flash flood disasters in 490 China from 2011 to 2019, China Flood & Drought Management, 30(9/10):22-25, DOI: 1673-9264 491 (2020) 09/10-22-04,2020. 492 Martina, M. L. V., Todini, E., and Libralon, A.: A Bayesian decision approach to rainfall thresholds 493 based flood warning, Natural Hazards and Earth System Sciences, 10, 413-426, DOI: 494 10.5194/hess-10-413-2006,2006. 495 Montesarchio, V., Lombardo, F., and Napolitano, F.: Rainfall thresholds and flood warning: An 496 operative case study, Natural Hazards and Earth System Sciences, 10, 447-455, DOI: 497 10.5194/nhess-9-135-2009,2009. 498 Merz, B., Kreibich, H., Schwarze, R., and Thieken, A.: Review article "assessment of economic flood 499 damage", Natural Hazards and Earth System Sciences, 10(8), 1697-1724, 500 DOI:10.5194/nhess-10-1697-2010,2010. 501 FAN Jianyong., SHAN jiusheng., GUAN Min.,et al.: Research on analysis and calculation method of 502 critical precipitation of mountain torrents in Jiangxi province, Meteorological Monthly, 503 38(9):1110-1114, DOI:10.7519/j.issn.1000-0526.2012.9.010,2012. 504 ZHANG Yaping., LIU De., LIAO Jun., et al.: A method for defining flood meteorological risk indices 505 of intermediate rivers based on hydrological simulation [J], Torrential Rain and Disasters, 31(4): 506 351-357, DOI:10.3969/j.issn.1004-9045.2012.04.009,2012. 507 Theule, J. I., Liebault, F., Loye, A., Laigle, D., and Jaboyedoff, M.: Sediment budget monitoring of 508 debris flow and bedload transport in the Manival Torrent, SE France, Natural Hazards and Earth 509 System Sciences, 12, 731–743, DOI: 10.5194/nhess-12-731-2012,2012.

Baogang Yang: Responsible for calculating critical rainfall thresholds for disaster causation using

510 PENG Tao., WANG Junchao., TANG Zhipeng., et al.: Analysis for calculating critical area rainfall on 511 different time scales in small and medium atchment based on hydrological simulation [J], 512 Torrential Rain and Disasters, 36(4): 365-372, DOI:10.3969/j.issn.1004-9045.2017.04.009,2017. 513 WANG Yuanjiang., JIANG Shanfu., LING Zhihan., et al.: Critical rainfall for flash flood warning 514 based on rainfall uncertainty [J], South-to-North Water Transfers and Water Science & 515 Technology, 22(1):90-98, DOI:10.13476/j.cnki.nsbdqk.2024.0011,2024. 516 Li Hanze., Zhang Jie., Yu Guohua., et al.: Research on Regional Early Warning of Flash Flood 517 Disasters in Small Watershed Based on Multi-source Early Warning Data Fusion, Advances in 518 Computer and Engineering Technology Research, 1 519 (2):406-418.10.61935/acetr.2.1.2024.p406,2024. 520 LU Yanyu., XIE Wusan., and TIAN Hong.: Analysis of critical flood causing rainfalls in medium and 521 small rivers based on hydrological model and statistical method, Journal of Natural Disasters, 522 25(3): 38-47, DOI:10.13577/j.jnd.2016.0305,2016. 523 ZHAO B., DAI Q., HAN D., et al.: Antecedent wetness and rainfall information in landslide threshold 524 definition[J], Hydrology and Earth System Sciences Discussions, 150:1-26, DOI: 525 10.5194/hess-2019-150,2019. 526 Teng Xiaomin., Zhang Xiaoxiao., Jiao Jiamin., et al.: Early warning index of flash flood disaster: a 527 case study of Shuyuan watershed in Qufu City, Water Science and Technology, 87(4), 892-909, 528 DOI:10.2166/wst.2023.016,2023. 529 Ewelina Janicka., Jolanta Kanclerz., Tropike Agaj., et al.: Comparison of Two Hydrological Models, 530 the HEC-HMS and Nash Models, for Runoff Estimation in Michałówka River, Sustainability, 15 531 (10):7959-7959, DOI:10.3390/su15107959,2023. 532 LIU Mingyan., SUN Fenghua., Zhou Xiaoyu., et al.: Threshold of Precipitation for Taizi River Basin 533 Flood Based on HBV Model, Desert and Oasis Meteorology, 15(5):109-115, 534 DOI:10.12057/j.issn.1002-0799.2021.05.015, 2021. Tang Guoan, and Yang Xin.: ArcGIS spatial analysis experiment course, 2<sup>nd</sup> ed, Beijing: Science Press, 535 536 289-292, 2012. 537 ZHANG Hongjiang., Kitahara Hikaru., Xie Mingshu., et al.: Study on roughness coefficient under the 538 conditions of several land utilization in the west of Shanxi province, Journal of Beijing Forestry University, 16(supp.4):86-91,1994. 539 540 Tong Xing. and Pei Yi.: Research on Relationship Between Surface Roughness and Hydraulic 541 Roughness, Hunan Agricultural Machinery, 542 38(3):35-37.DOI:10.3969/j.issn.1007-8320.2011.03.023. 2011. 543 GAN Yanjun., Li Nan., and Yang Mengfei.: Application of the SCS Model for Runoff Estimation in 544 Ungauged Basins, Yellow River, 32(5):30-31, DOI:10.3969/j.issn.1000-1379.2010.05.013, 2010. 545 Geomer.: Floodarea- Arcview extension for calculating flooded areas (User manual Version 2.4), 546 Heidelberg, 2003. 547 XU Jinxia., GUO Haiyan., DENG Guowei., et al.: Comparative study on rainfall threshold methods of 548 flash flood disaster: Take Mingshan River of Ya'an as an example, Journal of Natural Disasters, 549 33(3):89-99, DOI: 10.13577/j.jnd.2024.0308, 2024.

https://doi.org/10.5194/egusphere-2025-3833 Preprint. Discussion started: 15 October 2025 © Author(s) 2025. CC BY 4.0 License.

| 550 | WEI Wei., GAO Jing., YANG Jing., et al.: Study on the critical surface rainfall of flash floods at the  |
|-----|---------------------------------------------------------------------------------------------------------|
| 551 | South foot of Daqingshan in Hohhot Based on FloodArea model, Desert and Oasis Meteorology,              |
| 552 | 18(4):108-114, DOI:10.12057/j.issn.1002-0799.2024.04.015, 2024.                                         |
| 553 | JI Xingjie., LI Fengxiu., ZHU Yeyu., et al.: Determination of area rainfall threshold of flash flood in |
| 554 | the upper reaches of Luohe river of He'nan province, Journal of Meteorology and Environment,            |
| 555 | 31(6):43-50, DOI:10.3969 /j.issn.1673-503X.2015.06.006, 2015.                                           |
| 556 | HAPUARACHCHI H A P., WANG Q J., and PAGANO T C.:                                                        |
| 557 | A review of advances in flash flood forecasting, Hydrological Processes, 25(18): 2771-2784,             |
| 558 | DOI:10.1002/hyp.8040, 2011.                                                                             |
| 559 | Wang Zhiping.: Cause Analysis and Defense of Flash Flood in Lushi.Meteorological and                    |
| 560 | Environmental Sciences, 34(Suppl):138-140, DOI:10.16765/j.cnki.1673-7148.2011.s1.009, 2011.             |
| 561 | YU Baolong., LI Chao., LIU Liang., et al.: The warning indices analysis of rain flood relationship and  |
| 562 | critical rate of rainfall of Chongqing Tiaoshi river, Journal of Chengdu university of Information      |
| 563 | technology, 32(5):567-570, DOI:2096-1618(2017) 05-0567-04, 2017.                                        |
|     |                                                                                                         |