# Peer review of "Study on Critical Rainfall for Flash Flood Disasters in Small 1 Watersheds of the Three Gorges Reservoir Area: A Case Study of 2 Futian Small Watershed in Wushan County of Chongging 3 4 Qu Guo1, Qin Yang1\*, Jun Kang1\*, Yi Liu2, Baogang Yang1</su"

_EGUsphere, 2025_

## Author Comment (AC3)

Response to RC1

Dear Editors and Reviewers:

We would like to express our sincere gratitude to you for sparing your precious time to provide valuable and constructive comments on our manuscript entitled Study on Critical Rainfall for Flash Flood Disasters in Small Watersheds of the Three Gorges Reservoir Area: A Case Study of Futian Small Watershed in Wushan County, Chongqing (Manuscript ID: egusphere-2025-3833). These comments have played a crucial role in improving the quality of our manuscript and optimizing the research design.

We have carefully studied and analyzed all the reviewers' comments one by one, and clearly recognized that the original manuscript has several deficiencies in methodology description, research design rigor, result verification and logical flow. Adhering to the rigorous and realistic scientific research attitude, we have conducted a comprehensive and systematic revision of the manuscript. Below are our point-by-point responses to the reviewers' comments:

**Comment 1:**

In Section 2.2, the authors present a list of historical disasters... It is not clear whether this list was used solely for event identification and selection or whether it also served as validation for

the model results. For example, "The simulation results of this process are in good agreement with the actual situation...", the authors state that model results are in good agreement with actual conditions, yet the manuscript does not explicitly describe the evaluation methods or performance metrics used to support this claim.

Response: We highly appreciate your profound insights. The issues you pointed out are indeed major flaws in the original manuscript. We fully agree that the lack of clear validation methods and performance metrics has greatly compromised the reliability of our study.

To address this problem, we have made the following substantial revisions:

Introducing a "calibration-validation" framework: As clearly stated in Section 1.2 (Data Sources and Processing) of the revised manuscript, we divided 19 historical disaster events into 10 calibration events and 9 independent validation events. The calibration events were used to develop the model and determine preliminary thresholds, while the validation events were specifically employed to test the early-warning performance of these thresholds.

Adding a dedicated validation section: A comparative analysis of results derived from four calculation methods has been newly added

in Section 2.3 (Critical Rainfall Recommendation and Validation). We then recommended the critical rainfall and verified it using rainfall data from the 9 validation events. Validation results show that the recommended critical rainfall indicators successfully triggered early warnings for all 9 validation disaster events (i.e., rainfall amount for at least one duration exceeded the threshold), achieving a 100% hit rate. This quantitative result confirms the effectiveness of our method.

Revising vague expressions: Vague qualitative descriptions such as "in good agreement" have been revised in Section 2.1.1. We explicitly acknowledge the lack of measured inundation depth data for quantitative validation, and thus revised the description to "the simulated flood routing process... is basically consistent with post-disaster survey findings". We also emphasize that this is a qualitative validation, intended to illustrate that the model can reasonably reproduce the flash flood characteristics of the watershed rather than achieve precise numerical matching.

**Comment 2:**

Regarding the modeling approach, the study employs a CN-based hydrological model... a discussion on the accuracy and representativeness of the estimated parameters for the local

characteristics of the case study area is necessary. Furthermore, the manuscript notes that CN values depend on antecedent soil moisture conditions. However... it remains unclear how antecedent humidity conditions were accounted for in the regression analyses.

Response:Thank you for your attention to model parameterization, especially the critical technical detail of antecedent moisture handling. Your comments are highly professional and hit the nail on the head. We have supplemented and clarified this in detail in the revised manuscript as follows:

Adding a new subsection "Model Parameter Settings": In Section 1.3.2, we specifically discuss the sources and rationality of the key parameters of the FloodArea model (hydraulic roughness and SCS-CN values). We explain how these parameters are determined based on land use data and relevant research literature.

Clearly elaborating the handling method of antecedent moisture: Also in Section 1.3.2, we detail how antecedent soil moisture is quantitatively considered. We adopted the standard Antecedent Moisture Condition (AMC) classification method in the SCS-CN model. Specifically, based on the cumulative rainfall in the 5 days prior to each disaster event, we dynamically adjusted the CN value under the standard condition (AMC-II) to the CN value under the dry (AMC-I) or wet (AMC-III) condition. We calculated exclusive

CN spatial distribution maps reflecting the watershed moisture status at that time for each of the 10 flood events used for calibration.

Explaining how antecedent moisture is reflected in the regression: Since each simulation of the FloodArea model uses a specific CN value that reflects the antecedent moisture of that event, the simulated inundation depth results themselves have implicitly included the impact of antecedent moisture. Therefore, the subsequent "rainfall-inundation depth" regression relationship established (Figure 6) is actually built based on 10 sets of "input-output" relationships under different antecedent moisture conditions, thereby comprehensively considering the changes in antecedent moisture.

**Comment 3:**

Additionally, the models are presented as quadratic regression curves, while linear correlation coefficients are reported... which introduces some inconsistency that should be clarified.

Response: We highly appreciate your pointing out this methodological inconsistency. Your criticism is entirely justified, and reporting linear correlation coefficients for nonlinear regression is indeed misleading. We sincerely apologize for this oversight. The following corrections have been made in the revised manuscript:

In Sections 1.3.2 and 2.1.2, we revised the metric for evaluating

the goodness of fit of the regression model from "correlation coefficient" to the more appropriate Coefficient of Determination ($R^2$). $R^2$ can accurately measure the degree to which the quadratic polynomial regression equation explains the variance of the dependent variable (inundation depth).

We updated the result description in Section 2.1.2 to: "The results show that the coefficients of determination ($R^2$) of the quadratic polynomial regression models for all durations are above 0.7...", which ensures the consistency and scientificity of both the methodology and result reporting.

**Comment 4 & 5:**

Section 3.2, which describes the statistical approaches, would benefit from a clearer explanation of the data structures used for each method... Regarding the Single Station approach, it is also unclear whether using a spatially averaged precipitation series... still qualifies as a "single station" analysis or effectively represents a regionalized dataset.

Response: We sincerely appreciate your incisive comments on the statistical methodology section once again. The descriptions of data structure and method nomenclature in the original manuscript were indeed ambiguous and confusing. We have completely rewritten

Section 1.3.3 (Statistical Analysis Methods) to address these issues: Clarifying the data structure: We have clearly specified the sample data used for each statistical method.

Multi-station Extremum Averaging Method and Regional Critical Rainfall Method: We explicitly state that these two methods adopt rainfall data from the 10 calibration disaster events.

Probability Distribution Method: The sample dataset consists of the maximum rainfall sequences of different durations, derived from the maximum rainfall amounts recorded at each station across different sliding durations (1h, 2h, …, 24h) for the 10 calibration events.

Revising method names and definitions: Your query regarding the "single-station method" is entirely valid. It is indeed inappropriate to term the method "single-station" after averaging values from multiple stations. Therefore, we have renamed the original "Single-station Critical Rainfall Method" to Multi-station Extremum Averaging Method, and clearly defined its calculation steps in the method description: first extract the extreme values from all stations for each event, then calculate the average of these extreme values across different events. This ensures full consistency between the method name and its implementation.

Clarifying the data source for the Regional Method: We have

supplemented the description of the Regional Critical Rainfall Method with the statement: "This method only uses rainfall data from recorded disaster events for analysis", so as to eliminate ambiguity.

**Comment 6:**

Finally, in the Probability Distribution Method, the authors assess the goodness of fit of several theoretical distribution functions. Table 5 suggests that this comparison was based on the magnitude of the Kolmogorov–Smirnov (KS) statistic. However, the KS statistic... D-values from different distributions are not directly comparable. More appropriate model comparison criteria... should be used.

Response: We sincerely appreciate this extremely important professional comment on statistics. You are absolutely correct—directly comparing K-S statistic values from different distributions is statistically inappropriate. This was a serious academic error, for which we feel deeply remorseful, and we are grateful to you for helping us correct it. We have adopted your suggestion and made fundamental revisions to this section:

Adopting the AIC criterion: We abandoned the incorrect K-S value comparison in the original manuscript and instead used the academically recognized Akaike Information Criterion (AIC) as the

standard for model selection. A brief introduction to the principle of AIC has been added in Section 1.3.3.

Updating results and figures/tables: We recalculated the AIC values of all distribution functions and replaced the original table with an entirely new Table 5 (Comparison of AIC Values for Goodness of Fit of Different Probability Distribution Functions). The new results clearly show that except for the 1-hour duration, the Generalized Extreme Value (GEV) distribution yields the smallest AIC values for all other durations and is thus selected as the optimal fitting distribution.

This revision not only corrects the error but also significantly improves the scientificity and reliability of our research methodology. We once again express our admiration and gratitude for your rigorous academic spirit.

We would like to express our sincere gratitude again for the valuable comments provided by the reviewers.We look forward to your further review.

Sincerely yours

The Authors

December 24, 2025

---

## Author Comment (AC5)

Response to RC2

Dear Editors and Reviewers:

We would like to express our sincere gratitude to you for sparing your precious time to provide valuable and constructive comments on our manuscript entitled Study on Critical Rainfall for Flash Flood Disasters in Small Watersheds of the Three Gorges Reservoir Area: A Case Study of Futian Small Watershed in Wushan County, Chongqing (Manuscript ID: egusphere-2025-3833). These comments have played a crucial role in improving the quality of our manuscript and optimizing the research design.

We have carefully studied and analyzed all the reviewers' comments one by one, and clearly recognized that the original manuscript has several deficiencies in methodology description, research design rigor, result verification and logical flow. Adhering to the rigorous and realistic scientific research attitude, we have conducted a comprehensive and systematic revision of the manuscript. Below are our point-by-point responses to the reviewers' comments:

**Major Points:**

The aim of the study indicates that four methods will be compared. However, in the methodology section, only the Floor Area Model is described... How were the outcomes of the other

methods acquired? The authors do not justify how the Area Flood model was parameterized... I do not understand why 10 out of 19 flooded events were randomly selected... The regression analysis between inundation depth and cumulative area suddenly appears in the results... I cannot follow how the conclusions are related to the methodology and results. In the current status the study is hard to follow and understand.

Response:We highly appreciate your comments on the overall structure, methodological completeness, and logical flow of the manuscript. After in-depth reflection, we fully agree with your views that the original manuscript had serious flaws in these aspects, making it difficult for readers to understand. We have revised the structure and content of the entire manuscript.

Incomplete methodology: We have provided comprehensive and detailed descriptions of all four methods in Section 1.3 (Research Methods). Specifically, Section 1.3.2 elaborates on the hydrodynamic model method, while Section 1.3.3 systematically presents the principles, data sources, and calculation steps of the three statistical methods (Multi-station Extremum Averaging Method, Regional Critical Rainfall Method, and Probability Distribution Method).

Unclear model parameterization: A new subsection entitled

"Model Parameter Settings" has been added to Section 1.3.2. It details the basis for determining hydraulic roughness (Strickler coefficient) and runoff coefficient (SCS-CN), and specifically explains how to dynamically adjust CN values according to the antecedent moisture condition (AMC) of each event, thereby providing sufficient justification for the model parameterization process.

Unclear reasons for selecting 10 out of 19 events: The logic for event selection has been clearly explained in Section 1.2 (Data Sources and Processing). We divided the 19 events into 10 calibration events (for model construction and parameter calibration) and 9 validation events (for independent evaluation of model performance). This is a standard "calibration-validation" approach aimed at avoiding model overfitting and testing its generalization ability, which has been clearly stated in the manuscript.

Unclear purpose of regression analysis: In the "Critical Rainfall Determination Process" in Section 1.3.2, a specific step (③ Construct Regression Model) is included to elaborate on the purpose and method of regression analysis. That is, by establishing a quantitative relationship between "rainfall-inundation depth", the critical rainfall corresponding to a specific inundation depth (disaster level) is inversely derived. In this way, the regression analysis is

pre-explained in the methodology section rather than appearing abruptly in the results.

Confusing logical flow: The logical line of the entire manuscript has been reorganized to follow a clear thread: the introduction presents the research objectives→the methodology section elaborates on the "calibration-validation" design and the specific operation of the four methods→the results section first presents the calculation results of the four methods, then conducts comparative analysis to derive the recommended critical rainfall, followed by validation→the conclusion and discussion section summarizes the findings and reflects on the research. We believe the revised manuscript has a clear logic and is easy to understand.

**Minors:**

Figure 1. You should fill the blank space with the elevation map and outline the borders of the Futian Small watershed. Now it looks isolated.

Response:Thank you very much for your valuable suggestions on optimizing the charts. We have adopted your opinions as follows:

Revised Figure 1: We have redeveloped Figure 1. The new chart uses a Digital Elevation Model (DEM) as the base map, which intuitively displays the topographic relief characteristics of the study

area. It also clearly marks the watershed boundaries, river networks and relevant stations, eliminating the sense of isolation and significantly enriching the information content.

Table 1: This table could be more representative as a graphic showing the events and their duration.

Response:We sincerely appreciate the reviewer's thoughtful suggestion to present the information in Table 1 graphically. While we agree that visual representations can enhance interpretability, we have opted to retain the tabular format for two key reasons.

First, Table 1 includes multiple heterogeneous variables—such as casualty counts, economic losses, and affected crop areas—that vary substantially in scale and unit. Representing all these dimensions simultaneously in a single graphic would likely introduce visual clutter or risk misinterpretation.

Second, as outlined in the methodology section, the primary focus of this study is to establish the relationship between hydrological processes (i.e., precipitation and inundation depth) and critical rainfall thresholds. The disaster impact data in Table 1 serve mainly to support event selection and contextual validation rather than as core analytical variables. Nevertheless, they provide essential background on the severity and scope of each flood event, which strengthens the credibility of our threshold analysis.

For these reasons, we believe the current tabular format best preserves the completeness and clarity of this contextual information while avoiding unnecessary complexity in visualization.

L145: It doesn't explain how these coefficients will be used.

Response: Thank you for your valuable comments. We have explicitly added the following sentence at the beginning of Section 1.3.1:"Surface hydraulic roughness(expressed by the Manning-Strickler coefficient) and surface runoff coefficient (based on the SCS-CN method) are two key input parameters for simulating the surface runoff process in the FloodArea model." We have also provided explanations for these two key parameters.

L157: What about the description of the other methods?

Response: Thank you for your valuable comments. We have supplemented detailed descriptions of the three statistical methods in Section 1.3.3.

L159: You should add the reference.

Response: Thank you for your valuable comments. We have added the citation to the official technical document of its developer company when the model is first mentioned (Geomer, 2003).

L173:179: This is methodology. Please move it to the methods section.

Response: Thank you for your valuable comments. We have moved

this paragraph from Section 2.1 "FA Simulation" in the original manuscript to Section 1.3.2 "FloodArea Model Simulation Method" in the revised version and integrated it therein.

L157: You should describe how the methods will be compared.

Response: Thank you for your valuable comments. We have added a comparison of the calculation results of each method in Section 2.3. Then we adopted the "minimum value" principle to adjust and optimize the results of the four methods, selecting the minimum critical rainfall value among the four methods as the recommended flash flood critical rainfall for the corresponding duration. Subsequently, we used the rainfall data of 9 verification events to test the recommended critical rainfall. Verification results show that the recommended critical rainfall indicators successfully triggered early warnings for all 9 verified flood events (i.e., the rainfall amount of at least one duration exceeded the threshold), with a hit rate of 100%.

We would like to express our sincere gratitude again for the valuable comments provided by the reviewers.We look forward to your further review.

Sincerely yours

The Authors

December 24, 2025